# Information Retrieval and Extraction on COVID-19 Clinical Articles Using Graph Community Detection and Bio-BERT Embeddings

**Debasmita Das, Yatin Katyal, Janu Verma Shashank Dubey, Aakash Deep Singh,
Kushagra Agarwal, Sourojit Bhaduri, Rajesh Kumar Ranjan**
Mastercard AI Garage, Gurgaon, India
{firstname.secondname}@mastercard.com

## Abstract

In this paper, we present an information retrieval system on a corpus of scientific articles related to COVID-19. We build a similarity network on the articles where similarity is determined via shared citations and biological domain-specific sentence embeddings. Ego-splitting community detection on the article network is employed to cluster the articles and then the queries are matched with the clusters. Extractive summarization using BERT and PageRank methods is used to provide responses to the query. We also provide a Question-Answer bot on a small set of intents to demonstrate the efficacy of our model for an information extraction module.

## 1 Introduction

Novel coronavirus (COVID-19) has resulted in a pandemic in a short span of time owing to its quick transmission. A lot of scientific attention has been directed towards understanding the causes and impacts of the virus. Though this has resulted in a large amount of research articles being published every day, *extracting relevant information* (Asai et al., 2019) from such a huge pool of textual articles remains challenging. It is, thus, of particular importance to have systems that can retrieve relevant answers to queries. For example, it is useful to ask
*What is known about the transmission, incubation, and environmental stability of COVID-19 ?*

But finding information relevant to this query is quite challenging owing to the plethora of research articles being published, and the diversity, specificity of the query space. In this work, we address the problem of *extracting relevant answers* (Chen et al., 2017) from a corpus of clinical articles on COVID-19 in response to a query.

Information retrieval i.e. finding relevant documents in response to a query, is a standard problem with applications in web search engines e.g. Google, Bing etc. Models for retrieval systems rely heavily on *word-embeddings* (Mikolov et al., 2013) which provide a vector representation for every word in the corpus. The techniques developed for web search might not extend to the case of clinical articles since the distribution of words over these documents is quite different from the typical documents. There has been some work in building retrieval systems for scientific articles in a specific field e.g. biological or clinical papers, which are powered by domain specific *word-embeddings* (Mikolov et al., 2013) e.g. *BioBERT* (Lee et al., 2020). However, the direct use of *BioBERT* is not very optimal e.g. there might not even be an embedding of COVID-19. The domain of biological articles is still not specific enough to be able to used for our task. Thus, to build a useful information extraction system for COVID-19, it is important to fine-tune embeddings to align them to their distribution in COVID-19 related articles.

In this paper, we propose a system to extract information from a corpus of COVID-19 articles which is relevant to a *query* (Srihari and Li, 2000). Our approach has two main modules.

- **Graph-based Clustering:** This involves building a graph of the research articles in the corpus using the citations and textual similarity between them. Biological sentence vector embeddings are used to compute similarity[BioSentVec]. *Graph-based community detection* (Schaeffer, 2007) algorithms are employed to cluster the large number of documents into a relatively small number of clusters. We provide a detailed qualitative evaluation of the resulting clusters, and try to provide interpretable labels for the clusters. We find

best matching clusters to a query by computing the similarities of their BioSentVec vectors.

- **BERT-based Extractive Summarization:** This module extracts relevant sentences from the best-matched documents within the top clusters. *Contextual embeddings* (Si et al., 2019) of BERT-type trained on a corpus of biological articles are used to generate vectorial representation of sentences and the documents. Our output is a set of sentences from the summaries that are ranked in their degree of relevance to the query.

We demonstrate our model using **COVID-19 Open Research Dataset (CORD-19)** which was made available by the White House, Allen Institute for AI and a coalition of research groups. This data is available on Kaggle[1] as a part of their open research challenge (See Section 3.1). We also demonstrate the efficacy of our clustering and the summarization method by experimenting with a Question-Answer (QA) system where we provide precise answers to specific questions e.g. What is the incubation period of COVID-19 ? We provide evidence that our system can be employed, with minor modifications, on a much larger data to build a useful QA system or a chat-bot.

Concretely, we make following contributions:

1. Using graph-based clustering on a network of articles in the corpus.

2. Qualitative analysis of the clusters, and human-assisted labelling on the clusters.

3. Biological BERT based extractive summarization of the articles to find informative portions which are relevant to a query.

4. Proof-of-concept for a Question-Answer system on a limited set of intents.

The rest of this paper is divided as follows. In section 2 we will describe our method in details, the results are covered in section 3, evaluation and discussions are the part of section 4.

## 2 Method

In this section, we will discuss different components of our model. Starting point of out methodology is a graph of the articles.

---

[1]https://www.kaggle.com/allen-institute-for-ai/CORD-19-research-challenge

### 2.1 Construction of the graph

We build a *citation network* (Price, 1965) of the articles in the corpus where nodes corresponds to the papers in the corpus and the edges are determined by the citations of the papers. There are many ways a citation network can be constructed e.g. if paper A cities paper B, then there is a directed edge from A to B. We use the transversality of the citation relations to create edges i.e. if paper A and paper B both cite common papers then this is a signal that A and B are likely to be discussing similar topics.

In addition to the similarity of two papers in terms of their mutual citations, the semantic *similarity of the documents* (Lee et al., 2005) is also a valuable factor. Moreover, some articles might have only a few of their citations in the corpus, and some articles can have none of their citations in the corpus. Thus, we use semantic similarity between a pair of articles to add new edges and further enhance the coverage of network over the corpus. Word and *sentence embeddings* (Arora et al., 2016) have emerged as the standard way to obtain *semantic representation of textual documents* (Lau and Baldwin, 2016), where the documents are projected onto a low-dimensional space that preserves the semantic relationship. For this work, we use *BioSentVec* (Chen et al., 2019) which is trained on a corpus of about 30 million clinical and bio-medical research articles from the public databases - *PubMed* and *MIMIC-III*. *BioSentVec* provides *700-dimensional* ' sentence embeddings. We separately compute pairwise cosine similarity of article abstracts and the papers, and take their average as the *semantic similarity* (Muflikhah and Baharudin, 2009) between the papers. If this similarity of a pair of papers is greater than a threshold, we add an edge between them to the citation-based network.

Thus, we obtain a larger similarity-network of the papers containing un-directed edges. For simplicity of the discussion, we treat both types of edges i.e. citation-based and the semantic-based as indistinguishable and work with a homogeneous graph. The network, thus built, can have multiple edges between two nodes e.g. if they have multiple citations common and the edge can be formed via any of the shared citations, or if both types of edges are present. We ignore this multiplicity and consider at most one edge between any two nodes. It is possible to develop a *heterogeneous network* (Shi et al., 2014) of different edge-types, and edges

can be weighted according to the number of shared citations. We do not consider these approaches in this work.

## 2.2 Clustering of the papers

We employ *community detection* (Chen et al., 2011) on our graph of citation-based and semantic-based edges. Community detection is a useful technique to extract relationships between nodes in a complex graph. Nodes within a community are 'strongly' connected to each other than to those in different communities, and the nodes can be classified into *communities or modules* (Fortunato, 2010). For example, in a collaboration network of scientists, where nodes are scientists, edges corresponds to co-authorship, communities can indicate research areas. There is a plethora of community detection algorithms, each with their set of assumptions and workings. We will use community and cluster interchangeably.

For this task, we will use **ego-splitting** (Epasto et al., 2017) which provides a scalable and flexible community detection algorithm for complex networks. It employs local structures known as *ego-nets* which are the sub-graphs induced by the neighborhood of each node.

1. *Local ego-net clustering* involves construction of ego-nets for each node and then clustering of the ego-nets. For each cluster thus obtained, we add a new nodes (*personas*) which are same as the previous nodes but are now uniquely associated with a community. Then a new graph (*persona graph*) is constructed where there are multiple copies of nodes and the edges corresponds to the edges in the original network.

2. *Global network partitioning* involves the partitioning of the *persona graph* and mapping them back to the original graph.

This algorithm can be trained at different levels of resolutions - lower resolutions generate more granular clusters (higher number of clusters) and higher resolutions produce fewer clusters at a higher-level.

## 2.3 Mapping Queries to the Clusters

We next describe the method to map queries to clusters i.e. for any given query we find the clusters that are closely related to the query. We employ Bio-BERT embeddings to map the query and each document into dense vectors. Bio-BERT is a domain specific *BERT* (Devlin et al., 2018) (Bidirectional Encoder Representations from Transformers) for biomedical text mining, it is trained on corpus of PubMed and PMC full-text articles. It has been shown that Bio-BERT outperforms other approaches of embeddings as well as vanilla BERT on clinical data for variety of tasks e.g. *entity recognition* (Nadeau and Sekine, 2007), *relation extraction* (GuoDong et al., 2005), and QA system etc. This mapping is done in following steps:

1. Map title of each article in the corpus to a 768-dimensional vector using pre-trained Bio-BERT embeddings.

2. Obtain the Bio-BERT embedding for the given query.

3. Find top-40 most similar titles to the query in terms of their cosine similarities with the query.

4. This gives a distribution of cluster labels over the top-40 papers.

5. Based on a threshold on the similarity score or on the fraction of the top-40 papers in a cluster matching the query, we tag the query with a set of cluster labels.

This mapping helps in reducing the search space of the query and to retrieve more refined and focused results. It is worth noting here that the cluster assignment to the queries is done using only the titles, which might not capture the full relevance. But the assigned clusters do provide a direction and a smaller set of papers to explore further for better and faster search results.

Another purpose of this mapping of the query to the clusters is to purpose labels for each query in lieu of the supervised multi-label classification which is not possible due to the lack of the ground truth labels. More discussion in the 3.5 and 3.7.

## 2.4 Information Retrieval

We have reduced the set of possible articles that are relevant to a query as the union of the articles in the top-k clusters. Now, we will describe the process of retrieve articles that are best matched with the query. We again use the pre-trained Bio-BERT embeddings to obtain a vector representation of the whole document. This representation is different from the one used in that cluster mapping where

only title embedding is used. Also, we only consider the articles in the selected clusters, call this the *candidate set*. The Bio-BERT embedding of the query is used to compute its cosine similarity with the articles in the *candidate set*. Top-100 articles from the *candidate set* ranked by the cosine similarity with the query are selected to the returned in response to the query.

We also return a set of best matching sentences to the query which are deemed to be most informative. For this, we create a graph of sentences in the top-100 articles in terms of the cosine similarities of their Bio-BERT embeddings. The edges in this graph are weighted by the pairwise cosine similarities of the node sentences. Finally, the sentence nodes are ranked by their *PageRank* (Xing and Ghorbani, 2004) in this graph and top seven sentences are reported.

## 2.5 Question-Answer System

To explore the efficacy of our work for more refined information extraction, we experimented with a *Question-Answer bot* (Nomoto et al., 2004) which takes in a question and attempts of find the precise answer to it. For the input question, we employ our model to find relevant articles and passages which are most-likely to contain the answer. This question and the passage are then concatenated and fed to transformer to *BERT* (Devlin et al., 2018) type with pre-trained BioBERT embeddings as the inputs. The output layer is a sequence of the same length as input with a softmax layer that is trained to compute the probability of the corresponding input token to be the start and the end of the answer.

## 2.6 Extractive Summarization and Information Extraction

As a further enhancement and an application of the system, we provide *extractive summarization* (Padmakumar and Saran, 2016) of the best matched papers returned by the system. We attempt to produce a coherent summary of the papers by extracting important sentences from the paper. We used (Miller, 2019) approach for this task, which uses pre-trained *BERT* (Devlin et al., 2018) embeddings to obtain a sentence level embeddings and then *K-means clustering* (Wagstaff et al., 2001) of the sentences is performed. Finally, sentences closest to the centroid are selected.

## 3 Results and Discussions

In this section, we provide a discussion on the results and comment on the evaluation and broad utilization of this work.

## 3.1 Data

We used a corpus of scientific articles named **COVID-19 Open Research Dataset (CORD-19)** which was collected by the Allen Institute for AI and a coalition of research groups. Specifically, our motivation was the open research challenge hosted on Kaggle to build useful text mining tools to assist the medical community develop answers to high priority scientific questions. The *CORD-19* contains 134000 research articles, including 60000 full-text articles about *COVID-19, SARS, CoV-2* etc. Some important intents or tasks have been identified and there are multiple sub-tasks within each task. These represent a set of high importance topics and sub-topics for which relevant information to be retrieved from the given corpus.

As described in the Section 2.1 We built a network of the papers where each paper is represented as a node, and the edges between nodes implies that they either share a citation or the cosine similarity is greater than 0.9 for the BioSentVec embeddings of their abstracts and titles.

## 3.2 Clustering Results

We performed ego-splitting community detection algorithm on the article graph at various levels of resolution ranging from 0.001 to 1. The clustering that we report here was performed at the resolution of 0.3 to produce 661 clusters of non-uniform sizes consisting of around 38k papers.

We also attempt to provide human-understandable labels for some of the clusters. For each cluster, we select top-5 papers in each cluster using PageRank on the papers as nodes in the graph corresponding to the cluster. Using the keywords in these top articles provide us the candidates for labelling the clusters. These potential labels are then manually investigated against the cluster to refine the labels for it and to reduce noise in the label assignment. Some examples of clusters labels are as follows:

- Travel, Mass Gathering & Social Mixing during Epidemics (224 Research Papers).

- Clinical Management during Epidemics (223 Research Papers)

- Spread and Transmission of Viral Infections (3,347 Research Papers)

- Hospital Emergency Management (824 Research Papers)

- Social, Media/Newspapers & Political Impact on Viral Epidemics (95 Research Papers)

It must be noted that the labels do not faithfully match every single article in a cluster, but a substantial majority of the articles can be described by the assigned labels. We also do not claim to have 100% coverage since there a lot of clusters and we do not always have sufficient information to find consistent labels. Having a smaller set of labels helps in better bookkeeping. We plan to do a more careful study of the labels - automatically and manually - to further refine the results and increase the coverage over articles. We are also making public a set of articles with their labels. We hope that this set be used to study the articles related to COVID-19 in a supervised manner and to employ modern developments in NLP to develop techniques to help the community in various tasks.

### 3.3  Cluster Mapping

A sample of the cluster mapping results are shown in the Table 2. We take some examples of queries i.e. sub-tasks provided with the Kaggle competition and find their best-matching clusters via the procedure explained in Section 2.3. The results for all the sub-tasks are being made available.

### 3.4  Retrieval Results

A sample of results of the retrieval system for the sub-tasks provided with the Kaggle competition are shown in the Table 3. Consider the query sub-task : *Approaches to evaluate risk for enhanced disease & vaccinations starting after vaccination*, for which we find the best matching clusters as

1. Studies: Vaccine Development

2. Spread & Transmission of Viral Infections

From these clusters, the retrieval system finds the articles best matching to the sub-task as explained in 2.4.

1. Vaccines and Vaccination Practices: Key Food Systems to Sustainable Animal Production.

2. Canine Vaccination

3. Progress in Respiratory Virus Vaccine Development.

### 3.5  Discussion on Evaluation

Finally, we would like to address issues around the evaluation and applicability of this work. Since no ground truth data on articles matching the queries was provided, it was not possible to evaluate the system. We have thus no quantitative way to show superiority of our methods, neither do we claim any superiority. In fact, our motivation was to quickly prototype a retrieval system using modern advances in NLP like contextual embeddings e.g. BERT. We have used human intervention throughout this process both while building the model and for limited evaluation. We also propose potential evaluation methods in this situation.

We performed unsupervised clustering of the articles, evaluation of which is inherently difficult e.g. clustering is in the eyes of the beholder (Estivill-Castro, 2002). We do provide a qualitative study of the clusters by confirming that for most articles clusters within a cluster are 'more' similar to each other than to those in other clusters. It is possible to compute statistics like Silhouette coefficient, gap statistic etc. that provide a quantitative evaluation, but these statistics are not often useful or which of these should be used in not obvious, see e.g. Clustervision (Kwon et al., 2018). We also provide names/tags for the clusters based on finding top-papers in each clusters in terms of their PageRank values in a small graph and the keywords that figure prominently in these documents. Furthermore, we evaluated the tags by looking inside the clusters and comparing the papers against the proposed tags. Topic modeling e.g. LDA could be another approach to find tags for the articles and thus for the clusters. Our approach is much simpler and is also computationally efficient.

The information retrieval system that we proposed here works by finding articles that are best-matched to a query. We manually investigate the results for a set of queries i.e. sub-tasks in the Kaggle competition. First, the mapping of the queries to the clusters is done, then the best-matching documents are returned.

For the QA system, we provide short, precise answers to the questions. We make no claim on the correctness of the answers, and only restrict to

| Cluster | Example Papers | Journal | Published |
|---|---|---|---|
| Travel & Mass Gathering | 1. Mass gathering and globalization of respiratory pathogens during 2013 Hajj | Clinical Microbiology and Infection | 2015-06-30 |
| | 2. Travel implications of emerging coronavirus SARS and MERS-CoV | Travel Medicine & Infectious Disease | 2014-10-31 |
| | 3. Respiratory tract infections among French Hajj pilgrims from 2014 to 2017 | Sci Rep | 2019-11-28 |
| Studies: Vaccine Development | 1. Immunoinformatics and Vaccine Development: An Overview | Immunotargets Ther | 2020-02-26 |
| | 2. Immunization recommendations and safety & immunogenicity on the delayed vaccination of non-national immunization program for the coronavirus disease 2019 in China | Chinese Journal of Pediatrics | 2020-02-27 |
| Spread and Transmission of Viral Infections | 1. Prediction of COVID-19 Spreading Profiles in South Korea, Italy and Iran by Data-Driven Coding | medRxiv | 2020-03-10 |
| | 2. Importation and Human-to-Human Transmission of a Novel Coronavirus in Vietnam | New England Journal of Medicine | 2020-02-27 |
| | 3. Temperature significant change COVID-19 Transmission in 429 cities | - | 2020-02-25 |
| Impacts on Pregnancy | 1. From mice to women : the conundrum of immunity to infection during pregnancy | Journal of Reproductive Immunology | 2013-03-31 |
| | 2. Influenza and pneumonia in pregnancy | Clinics in Perinatology | 2005-09-30 |
| | 3. Pregnancy and perinatal outcomes of women with SARS | American Journal of Obstetrics and Gynecology | 2004-07-31 |
| Impact of Social Media | 1. Social media engagement analysis of U.S. Federal health agencies on Facebook | BMC Med Inform Decis Mak | 2017-04-21 |
| | 2. Social Media as a Sensor of Air Quality and Public Response in China | J Med Internet Res | 2015-03-26 |
| | 3. Scoping Review on Search Queries and Social Media for Disease Surveillance: A Chronology of Innovation height | J Med Internet Res | 2013-07-18 |

Table 1: Sample of clusters and corresponding papers.

| Sub-task | Clusters | No. of Documents |
|---|---|---|
| Approaches to evaluate risk for enhanced disease after vaccination | 1.Studies: Vaccine Development | 638 |
| | 2. Spread & Transmission of Viral Infections | 835 |
| Seasonality of Transmission | 1.Seasonality of Viral Infections | 138 |
| | 2. Spread & Transmission of Viral Infections | 835 |
| Age-adjusted mortality data for Acute Respiratory Distress Syndrome (ARDS) with or without other organ failure – particularly for viral etiologies | 1.Viral Infections - Studies | 3,347 |
| | 2.Hospital Emergency Management | 824 |
| | 3.Severe Pneumonia | 567 |

Table 2: Sub-tasks and their corresponding clusters

| Subtask | Top 3 Sentences | Title of the Paper | Journal |
|---------|-----------------|--------------------|---------|
| Approaches to evaluate risk for enhanced disease vaccination | 1. Cattle receive many vaccinations starting after 3 months of age after maternal immunity no longer interferes with vaccination by neutralizing the vaccine virus | Vaccines and Vaccination Practices: Key to Sustainable Animal Production | Encyclopedia of Agriculture and Food Systems |
| | 2. Although protection against most agents develops after routine vaccination programs, vaccines against some agents such as herpesvirus or Reovirus are not available. | Canine Vaccination | Veterinary Clinics of North America: Small Animal Practice |
| | 3. A realistic goal of immunization has to be a reduction of severe disease rather than induction of sterilizing immunity, similar to what has been achieved with rotavirus vaccines | Progress in Respiratory Virus Vaccine Development | Seminars in Respiratory and Critical Care Medicine |
| Co-infections (determine whether co-existing respiratory/viral infections make the virus more transmissible or virulent) and other co-morbidities | 1.The September epidemic of asthma-Observational studies have also been used to investigate the association of respiratory viruses with asthma morbidity | The Impact of Respiratory Viral Infection on Wheezing Illnesses & Asthma Exacerbations | Immunology and Allergy Clinics of North America |
| | 2.Their relatively short incubation times and efficient transmission via small droplets among comorbid patients highlight the need for better understanding of respiratory viral infections in hospital settings. | Laboratory-based surveillance of hospital-acquired respiratory virus infection in a tertiary care hospital | American Journal of Infection Control |
| | 3.However, such studies could not take into account possible episodes of mild or moderate illness that did not require inpatient medical care and could not address whether asymptomatic community spread played a role in the 2003 epidemic. | SARS-CoV Antibody Prevalence in all Hong Kong Patient Contacts | Emerg Infect Dis |

Table 3:  Sub-tasks & corresponding articles returned by the System

| Query | Answers | Confidence |
|---|---|---|
| Transmission Risks | 1. High Community Prevalence | 99.92 |
| | 2. VIral Infections | 99.84 |
| Animal Host to Human | 1. Virus | 99.79 |
| | 2. Pediculus lice | 99.64 |
| | 3. Anthropods | 98.974 |
| Risk Reduction Strategies | 1. Hygiene Measures | 98.235 |
| | 2. Antioxidant Vitamin Supplements | 97.746 |
| | 3. Quarantine | 97.471 |
| What are neonates risk? | 1. cardiac failure | 96.925 |
| | 2. Diarrhoea | 96.778 |
| | 3. Allergic Disorders | 96.424 |
| | 4. Serious Illness or Death | 95.439 |
| What are extrapulmonary manifestations of COVID-19? | 1. Orbital sinus bleeding | 92.95 |
| | 2. Invasive Devices | 90.21 |
| Coronavirus Survival | 1. 3 hours | 97.753 |
| | 3. 4 days on surfaces | 97.271 |

Table 4: Questions and Answers with Confidence Score

extracting the answers from the papers. Now it is possible that there are different answers reported to the same question in different papers.

### 3.6 Possible Evaluation

In lack of ground truth, a possible evaluation of retrieval system could be to perform a user-study on the *relevance* of the results to the query. This means measuring how relevant the results are to a query as ascertained by a set of unbiased users e.g. via Mechanical Turk. If we show titles and abstracts of top-k articles to the subjects, we can calculate the average number of articles marked relevant by them over a set of queries. This can be loosely interpreted as an estimate of the Precision k of the system. Since it is not known how many relevant articles there are in the corpus due to the lack of the notion of relevance, such an evaluation can not estimate recall value. It must be noted that such estimation is far from perfect since the variance across users and the queries are not factored in. Careful estimation of sample size is another point of contention. At least, such a system can help us reach a consensus notion of relevance and possibly building a small set of labeled data. Due to lack of time and complete clarity on the procedure, we have not performed evaluation of this kind, and restricted to our team-members investigating the results.

### 3.7 Further

We want to highlight that at various components of this system, we used human intervention to label data and tried to resolve some ambiguities. We are releasing these labeled pieces along with this paper. We hope this brings us closer to constructing a labelled data for various tasks e.g. classification of articles and queries into cluster categories, entity-recognition via top keywords, information retrieval & extraction, and a QA system.

## 4 Conclusion

In this work, we presented the problem of building an information retrieval system for scientific papers on COVID-19. This system, based on network analytical methods and modern developments in contextual word embeddings e.g. BERT, extracts articles and the sections therein relevant to a given query. We used human intervention in attempts to attach interpretable labels to the data e.g. articles, and queries. We also discussed challenges and possible avenues in evaluation of such a system in the lack of ground truth data.

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
