# OpenReview forum: "Information Retrieval and Extraction on COVID-19 Clinical Articles Using Graph Community Detection and Bio-BERT Embeddings"
_aclweb.org/ACL/2020/Workshop/NLP-COVID — NLP-COVID-2020 Abstractonly_

### Official Review · AnonReviewer1 · 2020-07-03
**An interesting search engine for CORD-19 but not properly evaluated.**

**Rating:** 4
**Confidence:** 3

**Review:**

The article describes a search engine dedicated to the CORD-19 corpus, a corpus focused on Covid-19 and released during the pandemic. The search engine relies on a similarity graph to find relevant documents for a given query. The nodes of the graph are the corpus articles and the edges are marking the existence of common citations between the articles or a high textual similarity between them. The ego-splitting algorithm was used to cluster the graph. Bio-BERT is heavily used to return, for a given a query, the most appropriate clusters, select the more relevant articles of the clusters selected, and the best matching sentences. Unfortunately, by lack of time and ground truth, the search engine was not quantitatively evaluated, only qualitatively evaluated.
Strength

    The article tackles an important problem and will provide a well-needed tool to help research. Can the authors give an estimation of the total time needed to implement and deploy the search engine? Is the search engine ready and available to be used by the community?
    The article is well written and overall easy to follow. Some typos.
    I found the idea to base the search engine on a similarity graph interesting and the intensive use of Bio-BERT should ensure the quality of the answers of the search engine.

Weakness

    The article has been submitted too quickly and should be revised before publication. The main limitation of the study, already acknowledged by the authors, is the lack of evaluation. I agree with the authors than without ground truth a quantitative evaluation is difficult to perform but a better qualitative evaluation is possible and should have been done before submission. In section 3.6 the authors described a possible way to estimate the precision of the search engine. Without resorting to Mechanical Turk, a simple double-blinded annotation should be sufficient to compute the statistic needed. To estimate the recall, a small set of articles could have been randomly selected and a manual evaluation of the clusters they have been assigned to, as well as the clusters they should have been assigned to, performed.
    There is no related work section and the article failed to compare the method proposed with the state-of-the-art methods in IR. I would also expect the quantitative evaluation to be performed on the results of a baseline system.
    The Q&A (and chatbot?) functionalities of the system are very briefly presented and could probably be discussed in another paper, focusing this paper on the search engine.

Details
Introduction
=> the first occurrence of BioSentVec in bracket is probably a reference missing, please double check
=> contribution 3 is difficult to read
=> contribution 4, what are the intents?
2.1
=> Was the unification of the references easy done? How the unification was performed, was the performance evaluated?
=> typo ‘cities’
=> if I understand correctly an edge is drawn between article A and B if A cites C and B cites C, a formal definition could  clarify the reading
=> The authors mentioned that for the simplicity of the discussion they consider the 2 types of links indistiguinshable but are they different in the computation? If yes, it has to be explained
=> typo: “if they have multiple citations in? common”
2.2
=> the ego-splitting algorithm should be explained in more detail. Without reading the reference, the idea of the algorithm is hard to understand in the current version
2.3
=> What are the cluster labels, they are defined yet
=> “is to purpose labels for each query in lieu of the supervised multi-label classification which is not possible due to the lack of the ground truth labels”, the sentence is unclear to me
2.6
=> a figure showing the full pipeline could be useful
3.1
=> the description of the data should be in the method section, not in the results. The authors should also refrain to use bold fonts
3.2
=> how 0.3 was chosen?
=> typo: “there a lot of clusters”
3.4
=> As it is currently phrased in the document, I am not sure if the authors are just using the data available on Kaggle or if they actually try to solve the shared-tasks, please precise
3.5
=> please, provide the numbers for the quantitative evaluation: how many clusters, how many articles were manually checked? Was the verification done double-blinded?

---

### Official Review · AnonReviewer4 · 2020-07-03
**An IR & IE engine for CORD-19 dataset works without the need for any labelled data**

**Rating:** 6
**Confidence:** 4

**Review:**

This paper presented a search engine that can perform both information retrieval and extraction for the COVID-19 Open Research Dataset.

The information retrieval component first converts the article collection into its graph representation. The graph is constructed by leveraging both Citation Network and semantic similarity between documents derived from sentence vectors produced by BioSentVec. A Community Detection algorithm Ego-splitting is then applied to the graph for identifying clusters of documents. When answering a given query, the system constructs a candidate set by gathering articles from the clusters of which document’s titles are most similar to the query in terms of BioBert embeddings. The system then compares the query vector with the BioBert vectors of entire articles in the candidate set and returns the most similar articles as final results.

The information extraction component has two different setups, Question Answering, and Summarisation. The former is done by using BioBERT for reading comprehension and the latter is done by K-means clustering on the sentence embeddings and extracting the centroids.

Strength

1. The paper presented a system that uses pre-trained and un-supervised models only. This is particularly meaningful as the time and resource for annotating data for COVID-19 related literature are very limited at the moment while the need for such systems is urgent.

2. The Kaggle shared task is used to aid the qualitative analysis to test the system performance when facing real-world information needs when there are limited resources for quantitative analysis.

Weaknesses

1. Although it has been mentioned in the introduction that the direct use of BioBERT might not be optimal, BioBERT is not fine-tuned by using the CORD-19 dataset.

2. No visualization to aid qualitative analysis. The system proposed for IR heavily depends on a graph-based clustering algorithm. However, there is no figure visualizing the derived map. The paper mentioned that the granularity of the map is controlled by a hyper-parameter, which is crucial to the entire system. But the choice of this parameter is not discussed and well-justified. Hence it might be interesting to visualize the map in different granularity and pick the one with the best cluster coherence.

3. Other than the use of BioBERT, it is not clear how the system is optimized for the CORD-19 dataset. It would be better if the system can leverage features that are specific to this context.


Details

Section 2
A flow chart of the entire pipeline would help in depicting the system at a high-level.

Section 2.1
How is the threshold for semantic edges selected?

Section 2.2
There are many concepts that the reader might not be familiar with here, such as ‘’ego-net’’ and ‘’ego-splitting’’, since the paper does not have a section dedicated for introducing related work, it would be clearer if more detail of these methods is provided.

Section 2.3 - 2.4
Here the system aims to reduce the set of candidates by filtering documents at cluster-level first. However, when doing cluster matching, the system still needs to compare the similarity between the BioBERT embeddings of the query and the title of each document. This should be fine for CORD-19. But for better scalability, it would be good to reduce the time complexity from the number of documents to the number of clusters.

Section 2.6
It is not clear whether this system uses a general BERT or BioBERT.

Section 3.1
It would be better to introduce the dataset earlier since it is supposed to be what the design of methodology based on.

Section 3.2
How are the 38k papers sampled from the CORD-19 dataset?

Section 3 (general)
Is the summarization system also evaluated?

---

### Official Review · AnonReviewer3 · 2020-07-06
**A Good Positioning or Pilot Study using Graph Community Detection and Bio-BERT Embedding for Topic Clustering and Query Matching with multiple potential applications**

**Rating:** 5
**Confidence:** 3

**Review:**

The paper is well-written with clarity and identification of gaps, limitation and future studies.

Strength: This is a good pilot study employing graph-based community detection and semantic similarity matching at sentence level using pre-trained bio-BERT embedding model for topic clustering and query matching.  It also describes experiments using the algorithms for information retrieval, document summarization, and question and answer bot.

Weakness: 1. Due to lack of groud-truth data sets, neither of the methods are evaluated quantitatively. 2. The topic clustering using only the paper title can be limiting as an article may have multiple topics.  3. The information retrieval system returning the relevant article based either on the paper title or sentence similarity is more of relevant document retrieval rather than information retrieval given the query.  This weakness is also reflected in the Q&A bot because returning an article or even a passage to a question is not the same as returning precise answer to the question as expected or desired by the user from Q&A bots. 4. Although the authors identified the limitation of pre-trained Bio-BERT model doesn't include COVID-19, the paper did not describe how it overcome this limitation.

However, the authors are trained in the field and familiar with the relevant literature.  Given the time, this study has good research and application potential.

---

### Decision · Program_Chairs · 2020-07-06

**Decision:**

Accept (Abstract only)

**Comment:**

The authors have presented a potentially interesting approach to retrieval in the context of the CORD-19 data set. However, the presentation of the work is not adequately grounded in prior research, specifically in information retrieval research, and the authors have not provided an evaluation of the methodology, on the CORD-19 data or on another data set. Evaluation data for the IR aspects of the work are now available from TREC-COVID (https://ir.nist.gov/covidSubmit/ ) utilising the CORD-19 data set.

While the paper itself is not quite ready for publication, the approach that the authors have outlined is interesting and we would like to include a presentation of the work in the workshop (Thursday 09 July, 17:30-21:30 PDT). Please prepare a 10-minute presentation for inclusion.

Thank you for submitting your work.

---

> ### Author Response · Authors · 2020-07-07
> **Thank You for the Feedback**
>
> Thanks for your valuable feedback. We find this decision very satisfactory.
> We agree with the suggestions to incorporate evaluation and also exploring the TREC challege for evaluation of the IR components.